# The COVID-19 Pandemic Impact on the Psychophysical Health of Post-Menopausal Women: A Cross-Sectional Study

**DOI:** 10.3390/medicina59061154

**Published:** 2023-06-15

**Authors:** Alessandra Lami, Giulia Giacomelli, Jacopo Lenzi, Stefania Alvisi, Renato Seracchioli, Maria Cristina Meriggiola

**Affiliations:** 1Gynecology and Physiopathology of Human Reproduction, IRCCS Azienda Ospedaliero-Universitaria di Bologna, Via Massarenti 13, 40138 Bologna, Italy; 2Department of Medical and Surgical Sciences, Alma Mater Studiorum University of Bologna, Via San Giacomo 12, 40126 Bologna, Italy; 3Department of Biomedical and Neuromotor Sciences, Alma Mater Studiorum University of Bologna, Via San Giacomo 12, 40126 Bologna, Italy

**Keywords:** menopause, COVID-19, menopausal hormonal therapy (HT), post-traumatic stress disorder (PTSD), MENQOL, depression

## Abstract

*Background and Objectives*: The aim of this study was to investigate lifestyle, menopausal symptoms, depression, post-traumatic stress disorder (PTSD) and sleep disorders in post-menopausal women throughout the first wave of the COVID-19 pandemic, including the impact of menopause hormonal therapy (HT). *Materials and Methods*: Post-menopausal women were given the following questionnaires: socio-demographic characteristics; lifestyle; history of COVID-19; menopause-specific quality of life (MENQOL), the first part of which refers to the pre-pandemic period (“pre COVID-19”) and the second refers to the current period (“during COVID-19”); Beck’s depression inventory (BDI); the impact of event scale-revised (IES-R); and the Pittsburgh sleep quality index (PSQI). *Results*: One hundred and twenty-six women completed all questionnaires, with a mean age of 55.5 ± 6.0 years. The mean menopause duration was 5.7 ± 5.6 years. Twenty-four women were taking HT. A significant mean weight gain, a reduction in physical activity (respectively *p* < 0.001) and worsened quality of romantic relationships (*p* = 0.001) were reported during the pandemic. Menopausal symptoms did not vary significantly throughout the pandemic; however, women taking menopausal HT had lower physical (*p* = 0.003) and sexual (*p* = 0.049) MENQOL domain scores, lower depressive symptoms (*p* = 0.039) and better romantic relationships (*p* = 0.008). *Conclusions*: The COVID-19 pandemic caused reduced physical activity, worsened food habits and weight gain in post-menopausal women. They also reported a high rate of severe–moderate PTSD and a negative influence on their romantic relationships. Menopausal HT seems to be a potential protective factor for sexual and physical status and for symptoms of depression.

## 1. Introduction

The Coronavirus 2019 (COVID-19) pandemic spread over all continents, registering more than 485 million cases worldwide [1]. In February 2020, Italy was the first European country to face the pandemic emergency [2].

The World Health Organization promoted the implementation of individual protection measures and the Italian government declared a state of emergency, imposing a full lockdown from March to May 2020 [3]. 

The psycho-social impact of the pandemic and the containment measures undertaken is considered a relevant public health issue. The onset of anxiety, stress and depression in populations that faced COVID-19 has been widely reported, especially in females, such as pregnant women [4,5]. Menopause is a delicate phase in the life of a woman, characterized by the insurgence of symptoms strongly influenced by conditions of stress [6]. We hypothesized that the current pandemic could have repercussions on patients who, even in normal situations, suffer physical symptoms from psychological factors [7,8].

Even though the psychological effects of the COVID-19 pandemic have been described in scientific literature, scant interest has been shown in post-menopausal women in particular. Little is known about the impact of COVID-19 on menopausal symptoms and on the psychological health of post-menopausal women. In our study, we investigated the effects of the pandemic on various psychological and physical aspects in post-menopausal women attending our outpatient clinic. In particular, we evaluated the onset of feelings of worry and anxiety related to the pandemic and compared the intensity of menopausal symptoms before and during the pandemic. We also investigated other aspects of the emotional response to the pandemic such as depression, post-traumatic stress disorder (PTSD) and sleep disorders, and whether lifestyles changes occurred in this population during lockdown. Analyses were stratified by the use of hormonal therapy (HT) to understand whether HT offered protection against psychological and/or physical deterioration in post-menopausal women during the pandemic.

## 2. Materials and Methods

### 2.1. Study Population

We enrolled consecutive post-menopausal women who attended the outpatient clinic of Gynecology and Physiopathology of Human Reproduction Unit at IRCCS of S. Orsola, Bologna, from May to December 2020. 

The study protocol was approved by our hospital Ethics Committee and written informed consent was obtained from all women in accordance with the 1975 Helsinki Declaration.

Each patient attending our clinic for a routine gynecological examination received information about the study. Those who consented to take part in our study received questionnaires in printed or electronic formats using a tablet (iPad), according to patient preference. Patients were asked to complete the questionnaires anonymously after the gynecological examination. The online questionnaires are available at Google forms link: https://docs.google.com/forms/d/1w6Uh2neweQ6yI0roYvtY1GOyBV-xJ_5W2kQSzVTZi10/edit?usp=sharing (accessed on 31 May 2023). (English version available in Appendix A).

Subjects were then divided into two groups according to the use or not of menopausal HT. 

### 2.2. Study Design

This was a cross-sectional study with retrospectively collected data as the enrolled women were asked to answer questions relating to symptoms reported before (from May 2019 to February 2020) and during the pandemic (from February 2020 up to the time of enrollment), with specific separate questionnaires entitled “Pre COVID-19” and “COVID-19”.

The inclusion criterion was a diagnosis of menopause (defined as amenorrhea for at least 12 months) [9]. Exclusion criteria were history of traumatic events other than the pandemic in the year before enrollment, cognition disorders and history of alcohol or substance abuse. 

### 2.3. Questionnaires 

Subjects answered a specific 36-item questionnaire (MP-nCoV19) to collect information regarding their socio-demographic characteristics, lifestyle before and during the pandemic, COVID-19 infection history, feelings of worry and anxiety and need for gynecological assistance during the pandemic. 

The presence and intensity of menopausal symptoms was assessed with the menopause-specific quality of life (MENQOL) questionnaire (“COVID19”), consisting of 29 items related to four areas: vasomotor (items 1–3), psycho-social (items 4–10), physical (items 11–26) and sexual (items 27–29). Symptoms may be present or absent; if present, the subject evaluates the related discomfort on a scale from zero (no discomfort) to six (substantial). Each MENQOL domain has a separate score ranging from one to eight [10,11]. A second questionnaire, identical to the previous one, was given to all subjects, asking them to answer with reference to the period before the pandemic (“Pre COVID19”).

The presence of depression was assessed by Beck’s depression inventory (BDI), a 21-item multiple-choice questionnaire. Each answer has a score from 0 to 3, with a total score ranging from 0 to 63 representing the grade of depression, with more serious symptoms of depression as the score increases [12,13].

Symptoms of PTSD were analyzed by the impact of event scale-revised (IES-R), a 15-item list. Subjects indicate how frequent the thoughts described in each item have been in the last seven days (never, rarely, sometimes and often). Each answer has a zero to four score, resulting in a total score ranging from zero to 88. The final score is related to the grade of PTSD, with more serious symptoms as the score increases and possible diagnosis of PTSD with a score over 24 [14].

We also used the Pittsburgh Sleep Quality Index (PSQI) to assess overall sleep quality, referring to the majority of nights of the previous month. The scale is generated by 19 individual items, referring to seven subcomponents which generate a global score. Each item has a zero to three Likert scale, where zero indicates “no problem” and three “severe problems”. The sum of these produces a global score ranging from zero to 21, with a global score of over five points indicating sleep disorders [15].

### 2.4. Statistical Analysis

We determined a minimum sample size of 122 by assuming a pre-pandemic mean MENQOL scoring of 3.5 ± 2.0 and a pandemic mean MENQOL scoring of 4.0 ± 1.9 (either vasomotor, psychosocial, physical or sexual) with correlation between paired observations of 0.5, power of 0.8, and significance level of 0.05 [16,17]. This calculation was based on a two-sample paired-means *t*-test, assuming that the two correlated samples are drawn from two normal populations and that the standard deviation of the differences within pairs is $\sqrt{\sigma_{1}^{2}+\sigma_{2}^{2}-2\rho \sigma_{1}^{2}\sigma_{2}^{2}}$, where $\sigma_{1}$ and $\sigma_{2}$ are the before-pandemic and during-pandemic group standard deviations, respectively, and ρ is the correlation between the paired measurements.

Continuous variables were summarized as mean ± standard deviation, while discrete and categorical variables were summarized as counts and percentages. Statistical significance in before/during differences was evaluated using the Wilcoxon matched-pairs signed-rank test, performing such evaluations on the entire sample and separately on HT versus non-HT patients. Crude comparisons between HT and non-HT patients were performed using the Wilcoxon rank-sum (Mann–Whitney) test or Fisher’s exact test, where appropriate, while proportional odds (ordered logistic) regression analysis was used to adjust for the confounding effects of differences in age structure between the two study groups. In secondary analysis, the Kruskal–Wallis test was used to compare BDI, IES-R and MENQOL scoring across different quality levels in the romantic relationships of subjects. Non-parametric tests were preferred over their parametric counterparts due to the low number of subjects under HT. The significance level was set at 0.05, and all tests were two-sided.

Sample size determination and data analysis was carried out using Stata 17 (StataCorp. 2021. Stata Statistical Software: Release 17. College Station, TX, USA: StataCorp LLC). 

## 3. Results

### 3.1. Study Population

The study was proposed to 160 women. Ten subjects refused for personal reasons. We enrolled 150 women from May 2020 to December 2020; twenty-four of these did not complete all the questionnaires and were excluded. One hundred and twenty-six women were included in the analysis.

The socio-demographic characteristics of the enrolled subjects are shown in Table 1. Of the 126 women enrolled, 24 were taking menopausal HT (HT-group). The mean duration of menopausal HT was 14 ± 7.5 months. Women using HT were significantly younger than those not taking HT (*p* = 0.015).

Arterial hypertension was present in 15.1% of women, 1.6% had diabetes, 10.3% had a previous cancer diagnosis, 2.4% had an immune system disorder and 19.8% had other comorbidities. 

### 3.2. Emotional Response to the COVID-19 Pandemic 

In our sample, 114 (90.5%) women had not experienced COVID-19 infection, only 1 had been in isolation and 11 (8.7%) had done a swab for medical assessment with negative results.

One hundred and five women had concerns about having reduced access to healthcare during the pandemic, with 38.9% stating that they were “very” or “extremely” worried. 

One hundred and seven women expressed some degree of anxiety during the pandemic, with 42% stating that they were “very” or “extremely” anxious. One hundred and twenty-three women were worried about their own and their family’s health, 67.4% of whom to a high degree (Table 2). 

A total of 38.9% of women expressed the need for gynecological help during the pandemic; however, only two subjects failed to access healthcare.

No difference between the HT and non-HT groups was detected in all emotional parameters except for the worry concerning reduced access to healthcare, which occurred to a greater extent in women taking HT (Table 2).

### 3.3. Lifestyle Changes before and during the COVID-19 Pandemic

One hundred and seventeen women maintained the same residence at the time of enrollment and during lockdown (Table 3).

Only 19.8% of women maintained their usual work regime before and during the pandemic, whereas 8.7% had to leave their jobs and 39.7% became smart workers (Table 3).

Most women significantly reduced their physical activity during the pandemic (*p* < 0.001 in the overall sample, Table 3) and reported changes in their diet during the pandemic; only 9.6% stated that they ate less than usual or healthier food, with the remainder stating that their food habits worsened (Table 3).

Lifestyle habits such as alcohol consumption, smoking and sleep duration did not change during the pandemic and were similar in all subjects regardless of HT use (Table 3).

Women taking HT were more physically active (*p* = 0.012). 

### 3.4. Physical Characteristics before and during the COVID-19 Pandemic

We recorded a mean weight increase of 1.2 ± 3.1 kg comparing the periods before and during the pandemic (*p* < 0.001), and a corresponding mean BMI increase (*p* < 0.001) (Table 3).

We found no significant difference in weight or BMI throughout the pandemic when comparing women taking HT and those not (Table 3).

### 3.5. Mental Health Parameters 

At the time of enrollment, 73 women (57.9%) did not report symptoms of depression whereas 32.5% reported minimal–mild symptoms and 8% moderate–severe symptoms of depression. The mean BDI score was 10.1 ± 8.0, which is in the range considered minimal depression (Table 4).

The percentage of women who did not report any depressive symptoms was significantly lower in women taking HT in comparison to those not taking HT (*p* = 0.039); in fact, no-one in the HT group reported moderate–severe depressive symptoms, whereas 10 women not taking HT (10%) reported severe or moderate symptoms and 35 (34%) mild or minimal symptoms (Table 4).

PTSD symptoms were found in varying degrees in almost half of the study population, with 39 (30.9%) subjects showing likely or severe grade PTSD, with a mean score of 25.9 ± 18.8, which is in the range considered as “a few signs” of PTSD (Table 4).

We did not find any significant influence of HT on PTSD, even though the IES-R score tended to be lower in the HT group. 

The majority of subjects reported poor sleep quality, with a mean PSQI score of 7.5 ± 3.8 (Table 4); however, the difference in sleep duration reported before and during the pandemic was not significant (Table 2).

There was no difference in sleep quality between the HT and non-HT groups (Table 4).

### 3.6. Quality of Life Related to Menopausal Symptoms and Romantic Relationships 

Of the 126 women enrolled, 109 were in a romantic relationship. Of these, the women in the HT group reported a better overall quality in their romantic relationships (*p* = 0.008) compared to those not taking HT. This remained unchanged during the pandemic (*p* = 0.750). The quality of romantic relationships did not seem to worsen in the HT group during the pandemic, whereas it worsened significantly in the overall study population and in those women not taking HT (*p* = 0.001) (Table 5).

When analyzing the MENQOL scores, we found a significant difference in the physical domain (*p* = 0.003), which was lower in the HT group during the pandemic compared to the non-HT group. We also found a slight but significant difference in the sexual domain (*p* = 0.049), which was lower in women taking HT (Table 5).

The other MENQOL domains remained unchanged throughout the pandemic across both groups. The same result was gained when referring to the perceived frequency of menopausal symptoms before and during the COVID-19 pandemic (Table 5).

The quality of romantic relationships did not significantly influence the presence of PTSD symptoms (*p* = 0.175), whereas it seemed to correlate with a lower BDI score (*p* = 0.011) and lower MENQOL psychosocial (*p* = 0.008), physical (*p* = 0.032) and sexual (*p* = 0.028) domain scores.

## 4. Discussion

In this study, we evaluated the psychological and the physical response of post-menopausal women to the COVID-19 pandemic using validated multiple-choice questionnaires. Questionnaires were administered at the end of the first wave of infection and most of the women enrolled had never been infected with COVID-19. We found a reduction of physical activity, weight increase and a high percentage (24.6%) of women with symptoms of PTSD, which are expected and relevant consequences of the pandemic and its subsequent restrictions.

We did not find a significant worsening of menopausal symptoms assessed via MENQOL questionnaires; however, women taking HT seemed to experience an improvement in the MENQOL physical and sexual domains compared to no change in the non-HT group, and they also reported fewer symptoms of depression.

Menopause is a delicate phase in the life of a woman, due to the onset of physical changes, new health issues and symptoms [6]. The most frequent symptoms are linked to estrogen deprivation, but external and psychological factors can also play a pivotal role [18,19,20,21]. The COVID-19 pandemic had a great impact on our everyday lives and many authors studied the psychological effects of the pandemic and subsequent confinement measures on the general population at the beginning [22,23,24] and throughout the pandemic [25]. The female population seems to have been affected the most [26].

However, specific information on how the pandemic affected post-menopausal women is limited, therefore we chose to focus on the physical and psychological health of this category of women during the COVID-19 emergency.

Among the main results of our study are the changes in physical activity and body weight, which are important results of the pandemic and are consistent with previous reports [27,28,29]. Confinement, the closure of gyms and swimming pools and mood impairment are the most common causes. Lum et al. [30] studied the physical activity of 164 women between the ages of 45–55 during the pandemic and found that women who met the UK Physical Activity Guidelines expressed lower depressive symptoms, lower psychological stress and better quality of life. Interestingly, there was no association with lower menopausal symptoms [31]. Similar findings are described by Coronado et al. [32], who found that peri- and post-menopausal women who engaged in physical and sexual activity had a higher quality of life and higher levels of resilience during the pandemic. In 2022, Vogel et al. [33] found that people who engaged in physical activity were more likely to manage stress and less likely to sleep or eat more during the pandemic. These findings should encourage us to consider the importance of physical activity as a coping strategy to improve not only physical but also psychological health, especially during crisis periods, also in post-menopausal women.

In our cohort we found no increase in smoking and alcohol consumption during the pandemic as was reported in other studies [27]; however, our study population contained few smokers and no heavy drinkers. In our opinion, our study group represents the Italian population in general [34,35].

Results regarding lifestyle are relevant also when considering the impact of depressive disorders, a sedentary situation and chronic inflammatory status related to stress and bad food habits, as well as the menopause itself, given that they are relevant cardiovascular risk factors in this population [36,37,38].

We also found a high impact on mental health parameters. In fact, our data show a relevant percentage (30.9%) of women with likely/severe PTSD symptoms (IES-R score > 33).

Other studies investigated the prevalence of PTSD in the general population after the COVID-19 pandemic. Most authors reported a higher prevalence of PTSD in women [39], especially in those over 60 years of age [40]. Another study carried out in Italy reported that 29.9% of women with endometriosis had a total IES-R score > 33, which indicates a probable diagnosis of PTSD [41] and described a correlation between a higher IES-R score and older age. Post-menopausal women have increased cardiovascular and epinephrine responses to mental stress compared to pre-menopausal women [42] and the lower estrogens/progesterone ratio correlates negatively with a higher frequency of involuntary distressing thoughts [43]. Liu et al. [22] studied 235 people living in Wuhan at the start of the COVID-19 pandemic and found a prevalence of PTSD of only seven percent, but with a higher prevalence in females versus males. The lower prevalence of PTSD in this study could be ascribed to many factors, including the fact that Liu et al. conducted their study at the very beginning of the epidemic, whereas we enrolled women when the virus had spread globally and its consequences were already well known. This difference may also be due to cultural factors and different scales used to assess PTSD.

On the other hand, Sievert et al. studied a population of women with a mean age of 51.1 years before and during the pandemic and did not find any difference in perceived stress [44]. These mixed results could be related to the different time of evaluation throughout the pandemic but also to specific cultural and geographic parameters. It would be interesting to investigate this topic with larger studies to evaluate stress perception among different populations.

In our cohort, the prevalence of moderate/severe symptoms of depression during the first COVID-19 wave was low (5.6% and 2.4%, respectively), whereas 57.9% of women reported no symptoms. This differs from the results of Varma et al. [45], who found moderate symptoms of depression in 39% of 1653 adults (mean age 42.9 years ± 13.6) from 63 countries, while Ran et al. [46] enrolled 1770 Chinese citizens (mean age 28.7 years ± 10.6) and found a depression prevalence of 47.1%. However, our findings may be in line with previous studies, which suggest that elderly people are more resilient than the younger population [47], with lower rates of psychiatric/psychological symptoms, potentially due to complex experiences during their longer lives [48]. A potential bias could be that only a few women had had direct experience of COVID-19. Nowadays, the number of contacts has risen exponentially and could affect the psychological wellbeing of women very differently.

Both the MENQOL and the PSQI did not highlight a significant difference in sleep duration before and during the pandemic, although 20 women out of 126 stated that they slept less during lockdown. Most women in our study reported suffering from poor quality of sleep before the pandemic, which may be related to the menopausal status itself [49]. Gargiulo et al. [50] reported that women had worse sleep quality compared to men during the pandemic [51].

The primary hypothesis of our study was that the pandemic and its related stressful conditions could affect menopausal symptoms; therefore, we compared the presence and intensity of these symptoms in our cohort before and during the pandemic via two MENQOL questionnaires specific to each period of time.

We found that most of the MENQOL domains did not significantly change at the end of the first wave of the pandemic, even though we did expect some worsening in at least the psycho-social domain as a consequence of the distress caused by the pandemic and of the limitations in everyday lives [4,52]. Sievert et al. also [44] found no differences in the frequency of menopausal symptoms such as aches/stiffness in joints, irritability and hot flashes before and during the pandemic.

This result differs from those of Monterrosa-Blanco, who [53] found that Colombian post-menopausal women who suffered more loneliness during the pandemic also suffered more menopause-related urogenital symptoms.

Although we found no significant variation in the MENQOL sexual domain throughout the pandemic, our cohort reported a worsening in the quality of their romantic relationships. This may be due to a variety of factors. The lockdown may have led to longer cohabitation time, which could have been negatively affected by the confinement measures and the situation of alarm. Schiavi et al. [54] studied the sexual activity of 89 Italian women aged 28–50 and found that the frequency of sexual intercourse decreased significantly despite the longer cohabitation time. Furthermore, the sexual function, measured by FSFI and FSDS, showed a significant worsening. A possible, or at least partial, consequence of this is the decline in birth rates during the first year of pandemic [55].

Subjects with a better quality of romantic relationship showed significantly lower BDI (*p* = 0.011) and MENQOL psychosocial (*p* = 0.008), physical (*p* = 0.032) and sexual (*p* = 0.028) domains [32]. These data highlight the importance of personal interactions in coping with stressful situations such as COVID-19.

Interestingly, women in the HT group had a significantly lower prevalence of depression symptoms, lower physical and sexual MENQOL domain scores, and reported better quality of romantic relationships during the pandemic. This may be partly due to the younger age of this population; however, we can also consider that HT improves general wellbeing [49] and sexual function [56]. On the other hand, the perceived frequency of menopause-related symptoms did not differ in our two groups. This may be because symptomatic women are more often prescribed HT in comparison to less symptomatic women and to the limited number of women taking HT.

We found no significant variation in the prevalence of PTSD or sleep disorders between the two groups. It is known that the HPA axis changes during menopause and may enhance the effects of stress. Post-menopausal women not taking HT tend to have greater cortisol response to acute stress than age-matched men and younger adults, and longer HT use seems to determine a lower cortisol response to stress [57]. In our cohort, the number women taking HT is low and the mean HT duration may explain the blunt effect on these symptoms.

The strengths of this study are that it analyzes how menopause-related symptoms could have been affected by the pandemic, and we collected data with validated questionnaires investigating many different psychological and physical aspects. Furthermore, it provides socio-demographic information regarding Italian women facing menopause and the first wave of the COVID-19 pandemic. We found that most women who needed gynecological assistance during the pandemic managed to obtain it, indicating that Italian hospitals guaranteed adequate support to patients and the pandemic did not significantly affect gynecological care as had happened in other contexts [58,59].

The main limitation of this study is the possibility of recall bias, since the retrospective compilation of the questionnaires referring to pre-COVID-19 may have led to erroneous or incomplete responses due to memory lapses. Another limitation is the small number of subjects taking menopause HT, although we believe this reflects the general Italian menopausal population [60,61,62].

Furthermore, we collected data at the end of the first wave of the pandemic only and not during or after the following waves; nor did we consider the effects of vaccines.

## 5. Conclusions

In conclusion, our data show that the COVID-19 pandemic significantly affected everyday life, being responsible for negative influences on the psycho-social spheres and romantic relationships of post-menopausal women. Other important changes are the lower engagement in physical activity, worsening of food habits and weight gain, which are also reported in other populations. In our study sample, we also found a high rate of severe to moderate PTSD.

HT seemed to be a potential protective factor, possibly providing a better physical and sexual status, with a positive impact on the risk of developing depression symptoms as well.

This study focuses our attention on the psycho-physical wellbeing of post-menopausal women, encouraging physicians to develop a holistic approach to them. The COVID-19 pandemic opened a brand-new scenario for public health and social life, and the health of post-menopausal women should not be neglected.

## 6. Patents

This section is not mandatory but may be added if there are patents resulting from the work reported in this manuscript.

## Figures and Tables

**Table 1 medicina-59-01154-t001:** Socio-demographic characteristics of the study population, overall and by use of HT.

Characteristics	All	HT	No HT	*p*-Value between Groups
	(n = 126)	(n = 24)	(n = 102)	
Residence				0.071
Metropolitan city of Bologna	83 (65.9%)	19 (79.2%)	64 (62.7%)	
Emilia-Romagna	15 (11.9%)	4 (16.7%)	11 (10.8%)	
Other regions of Italy	28 (22.2%)	1 (4.2%)	27 (26.5%)	
Age, y [mean ± SD]	55.5 ± 6.0	53.0 ± 4.6	56.1 ± 6.1	0.015 *
Menopause duration, y [mean ± SD]	5.7 ± 5.6	6.3 ± 4.1	5.5 ± 5.9	0.159
Marital status				0.381
Single	8 (6.3%)	2 (8.3%)	6 (5.9%)	
Married	79 (62.7%)	13 (54.2%)	66 (64.7%)	
Cohabiting	11 (8.7%)	2 (8.3%)	9 (8.8%)	
Divorced	22 (17.5%)	7 (29.2%)	15 (14.7%)	
Widowed	6 (4.8%)	0 (0.0%)	6 (5.9%)	
Occupation				0.406
Retired	18 (14.3%)	1 (4.2%)	17 (16.7%)	
Steady job	89 (70.6%)	19 (79.2%)	70 (68.6%)	
Occasional worker	6 (4.8%)	2 (8.3%)	4 (3.9%)	
Layoff	3 (2.4%)	0 (0.0%)	3 (2.9%)	
Unemployed	10 (7.9%)	2 (8.3%)	8 (7.8%)	
Parity				0.139
0	34 (27.0%)	10 (41.7%)	24 (23.5%)	
1	73 (57.9%)	10 (41.7%)	63 (61.8%)	
2–3	19 (15.1%)	4 (16.7%)	15 (14.7%)	

* *p*-value ≤ 0.05; SD, standard deviation.

**Table 2 medicina-59-01154-t002:** Emotional response to COVID-19 pandemic, overall and by use of HT.

Responses	All	HT	No HT	*p*-Value between Groups
	(n = 126)	(n = 24)	(n = 102)	
Worried about not having access to healthcare				0.008 *
No	21 (16.7%)	8 (33.3%)	13 (12.7%)	
Little	18 (14.3%)	0 (0.0%)	18 (17.6%)	
Somewhat	38 (30.2%)	10 (41.7%)	28 (27.5%)	
Very much	30 (23.8%)	5 (20.8%)	25 (24.5%)	
Extremely	19 (15.1%)	1 (4.2%)	18 (17.6%)	
Anxious				0.270
No	19 (15.1%)	4 (16.7%)	15 (14.7%)	
Little	18 (14.3%)	3 (12.5%)	15 (14.7%)	
Somewhat	36 (28.6%)	6 (25.0%)	30 (29.4%)	
Very much	28 (22.2%)	9 (37.5%)	19 (18.6%)	
Extremely	25 (19.8%)	2 (8.3%)	23 (22.5%)	
Worried about own health and family				0.216
No	3 (2.4%)	2 (8.3%)	1 (1.0%)	
Little	16 (12.7%)	1 (4.2%)	15 (14.7%)	
Somewhat	22 (17.5%)	4 (16.7%)	18 (17.6%)	
Very much	44 (34.9%)	8 (33.3%)	36 (35.3%)	
Extremely	41 (32.5%)	9 (37.5%)	32 (31.4%)	

* *p*-value ≤ 0.05.

**Table 3 medicina-59-01154-t003:** Lifestyle changes and physical characteristics before and during the COVID-19 pandemic, overall and by use of HT.

Lifestyle Changes during the Pandemic	All	HT	No HT	Two-Sample *p*-Value	Adj. 2-Sample
	(n = 126)	(n = 24)	(n = 102)		*p*-Value
Work				0.797	0.807
Same as before	25 (19.8%)	3 (12.5%)	22 (21.6%)		
More than before	24 (19.0%)	5 (20.8%)	19 (18.6%)		
Smart working	50 (39.7%)	12 (50.0%)	38 (37.3%)		
Less than before	8 (6.3%)	2 (8.3%)	6 (5.9%)		
Interrupted	11 (8.7%)	1 (4.2%)	10 (9.8%)		
Unspecified	8 (6.3%)	1 (4.2%)	7 (6.9%)		
Workout frequency				0.151	0.209
Less than before	37 (29.4%)	11 (45.8%)	26 (25.5%)		
Same as before	78 (61.9%)	12 (50.0%)	66 (64.7%)		
More than before	11 (8.7%)	1 (4.2%)	10 (9.8%)		
Paired *p*-value	<0.001 *	0.006 *	0.011 *		
Changes in diet					
Eating without schedule	15 (11.9%)	3 (12.5%)	12 (11.8%)	1.000	0.922
Eating more food	6 (4.8%)	1 (4.2%)	5 (4.9%)	1.000	0.955
Eating less food	5 (4.0%)	1 (4.2%)	4 (3.9%)	1.000	0.934
Eating unhealthier food	7 (5.6%)	2 (8.3%)	5 (4.9%)	0.617	0.594
Eating healthier food	7 (5.6%)	1 (4.2%)	6 (5.9%)	1.000	0.564
Ordering takeaways more often	26 (20.6%)	3 (12.5%)	23 (22.5%)	0.402	0.282
Cooking more often	6 (4.8%)	0 (0.0%)	6 (5.9%)	0.594	0.369
Alcohol consumption				0.891	0.689
Less than before	7 (5.6%)	1 (4.2%)	6 (5.9%)		
Same as before	106 (84.1%)	20 (83.3%)	86 (84.3%)		
More than before	13 (10.3%)	3 (12.5%)	10 (9.8%)		
Paired *p*-value	0.223	0.625	0.380		
Cigarette smoking				0.346	0.997
Less than before	1 (0.8%)	1 (4.2%)	0 (0.0%)		
Same as before	124 (98.4%)	23 (95.8%)	101 (99.0%)		
More than before	1 (0.8%)	0 (0.0%)	1 (1.0%)		
Paired *p*-value	1.000	1.000	1.000		
Sleep duration				0.315	0.388
Less than before	20 (15.9%)	4 (16.7%)	16 (15.7%)		
Same as before	96 (76.2%)	20 (83.3%)	76 (74.5%)		
More than before	10 (7.9%)	0 (0.0%)	10 (9.8%)		
Paired *p*-value	0.099	0.125	0.327		
Change of residence during the pandemic	9 (7.1%)	1 (4.2%)	8 (7.8%)	1.000	0.442
Weight, kg				0.867	0.961
During	65.5 ± 12.8	63.2 ± 9.6	66.1 ± 13.4		
Before	64.4 ± 12.1	62.2 ± 9.8	64.9 ± 12.5		
Delta	1.2 ± 3.1	1.0 ± 3.1	1.2 ± 3.1		
Paired *p*-value	<0.001 *	0.060	<0.001 *		
Body mass index, kg/m²				0.918	0.987
During	24.4 ± 4.6	23.6 ± 3.8	24.6 ± 4.8		
Before	24.0 ± 4.4	23.2 ± 3.9	24.2 ± 4.5		
Delta	0.4 ± 1.2	0.4 ± 1.1	0.4 ± 1.2		
Paired *p*-value	<0.001 *	0.062	<0.001 *		

* *p*-value ≤ 0.05; *Notes:* Paired *p*-values denote whether there is a significant before/during change; two-sample *p*-values denote whether there is a significant difference in before/during changes between the two study groups; adjusted two-sample *p*-values make the same assessment by adjusting for the confounding effects of differences in age structure between the two study groups being compared (see Table 1).

**Table 4 medicina-59-01154-t004:** Mental health parameters of the study sample at the time of the pandemic period, overall and by use of HT.

Mental Health Parameter	All	HT	No HT	*p*-Value between Groups
	(n = 126)	(n = 24)	(n = 102)	
Beck’s Depression Inventory (BDI) score [mean ± SD]	10.1 ± 8.0	8.1 ± 6.0	10.6 ± 8.3	0.266
Beck’s Depression Inventory (BDI) range scores [n, %]				0.039 *
No depression (≤10)	73 (57.9%)	16 (66.7%)	57 (55.9%)	
Minimal depression (11–16)	26 (20.6%)	2 (8.3%)	24 (23.5%)	
Mild depression (17–20)	15 (11.9%)	4 (16.7%)	11 (10.8%)	
Moderate depression (21–30)	7 (5.6%)	0 (0.0%)	7 (6.9%)	
Severe depression (>30)	3 (2.4%)	0 (0.0%)	3 (2.9%)	
Unspecified	2 (1.6%)	2 (8.3%)	0 (0.0%)	
PTSD (from IES-R) score [mean ± SD]	25.9 ± 18.8	19.0 ± 14.7	27.5 ± 19.4	0.076
PTSD (from IES-R) range scores [n, %]				0.475
No (<24)	66 (52.4%)	16 (66.7%)	50 (49.0%)	
A few signs (24–32)	13 (10.3%)	2 (8.3%)	11 (10.8%)	
Likely (33–37)	8 (6.3%)	1 (4.2%)	7 (6.9%)	
Severe (>37)	31 (24.6%)	3 (12.5%)	28 (27.5%)	
Unspecified	8 (6.3%)	2 (8.3%)	6 (5.9%)	
Pittsburgh Sleep Quality Index (PSQI) score [mean ± SD]	7.5 ± 3.8	7.1 ± 3.4	7.6 ± 3.9	0.638
Pittsburgh Sleep Quality Index (PSQI) range scores [n, %]				0.856
Good (<5)	34 (27.0%)	6 (25.0%)	28 (27.4%)	
Poor (≥5)	91 (72.2%)	18 (75.0%)	73 (71.6%)	
Unspecified	1 (0.8%)	0 (0.0%)	1 (1.0%)	

* *p*-value ≤ 0.05; N, number of cases; %, percentage; SD, standard deviation; PTSD, post-traumatic stress disorder; IES-R, impact of event scale–revised.

**Table 5 medicina-59-01154-t005:** Quality of life related to menopausal symptoms and romantic relationships before and during the COVID-19 pandemic, overall and by use of HT.

Changes during the Pandemic	All	HT	No HT	Two-Sample *p*-Value	Adj. 2-Sample
					*p*-Value
Quality of romantic relationships †	(n =109)	(n = 22)	(n = 87)	0.474	0.316
Worse than before	22 (17.5%)	3 (12.5%)	19 (18.6%)		
Same as before	81 (64.3%)	14 (58.3%)	67 (65.7%)		
Better than before	6 (4.8%)	2 (8.3%)	4 (3.9%)		
Paired *p*-value	0.001 *	0.750	0.001 *		
Frequency of menopause-related symptoms	(n = 126)	(n = 24)	(n = 102)	0.331	0.162
Less than before	6 (4.8%)	2 (8.3%)	4 (3.9%)		
Same as before	106 (84.1%)	21 (87.5%)	85 (83.3%)		
More than before	14 (11.1%)	1 (4.2%)	13 (12.7%)		
Paired *p*-value	0.114	1.000	0.060		
MENQOL vasomotor	(n = 126)	(n = 24)	(n = 102)	0.647	0.617
During	3.3 ± 2.1	2.8 ± 2.0	3.4 ± 2.1		
Before	3.3 ± 2.0	3.1 ± 2.2	3.3 ± 2.0		
Delta	0.0 ± 1.1	−0.4 ± 1.5	0.0 ± 1.0		
Paired *p*-value	0.152	0.246	0.322		
MENQOL psychosocial	(n = 126)	(n = 24)	(n = 102)	0.130	0.059
During	4.0 ± 1.9	3.6 ± 1.7	4.2 ± 1.9		
Before	3.9 ± 1.7	3.7 ± 1.6	3.9 ± 1.7		
Delta	0.2 ± 1.0	−0.1 ± 0.8	0.2 ± 1.0		
Paired *p*-value	0.436	0.406	0.191		
MENQOL physical	(n = 126)	(n = 24)	(n = 102)	0.003 *	0.008 *
During	3.6 ± 1.5	3.2 ± 1.2	3.7 ± 1.5		
Before	3.6 ± 1.5	3.5 ± 1.4	3.6 ± 1.5		
Delta	0.0 ± 0.6	−0.3 ± 0.6	0.1 ± 0.6		
Paired *p*-value	0.760	0.007 *	0.108		
MENQOL sexual	(n = 126)	(n = 24)	(n = 102)	0.049 *	0.030 *
During	4.6 ± 2.3	3.7 ± 2.2	4.8 ± 2.3		
Before	4.5 ± 2.3	4.0 ± 2.5	4.6 ± 2.2		
Delta	0.1 ± 1.0	−0.3 ± 0.9	0.2 ± 1.0		
Paired *p*-value	0.908	0.141	0.462		

* *p*-value ≤ 0.05; ^†^ Available for 109 subjects; *Notes:* Paired *p*-values denote whether there is a significant before/during change; two-sample *p*-values denote whether there is a significant difference in before/during changes between the two study groups; adjusted two-sample *p*-values make the same assessment by adjusting for the confounding effects of differences in age structure between the two study groups being compared (see Table 1).

## Data Availability

The data supporting the findings of this study are available from the corresponding author, A.L., upon reasonable request.

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
