# Peer review of "The COVID-19 Pandemic Impact on the Psychophysical Health of Post-Menopausal Women: A Cross-Sectional Study"

_medicina, 2023, doi:10.3390/medicina59061154_

Round 1
Reviewer 1 Report
The authors of this paper discovered lifestyle, menopausal symptoms, depression, PTSD and sleep disorders in post-menopausal women throughout the first COVID-19 pandemic. I was pleased to read this original paper. The authors have well substantieted the aim of the study and why they choose this particular cohort. The methods were described comprehensivly and according to the design of the study. The paper is well written and all the references were used correctly.
I would like reccomend to author to add the information about design in title or abstract. It will improve understanding of the study when reviewing the abstract. Also It would be better to unify the aim of the study in the abstarct and in the introduction.
Author Response
- Point 1: The authors of this paper discovered lifestyle, menopausal symptoms, depression, PTSD and sleep disorders in post-menopausal women throughout the first COVID-19 pandemic. I was pleased to read this original paper. The authors have well substantiated the aim of the study and why they choose this particular cohort. The methods were described comprehensively and according to the design of the study. The paper is well written and all the references were used correctly.
Response 1: Thank you for your kind and enthusiastic comment. We are glad to know you appreciate our work.
- Point 2: I would like recommend to author to add the information about design in title or abstract. It will improve understanding of the study when reviewing the abstract. Also, it would be better to unify the aim of the study in the abstract and in the introduction.
Response 2: Thank you for your suggestion. We have now declared the study design in the title (line 3), and have also harmonized the study aim stated in the abstract and the introduction (lines 68-71).
Reviewer 2 Report
1-Calculation of sample size needs to be revised. Which formula or program was used for sample size calculation?
2-Study design and setting need to be thoroughly described
3-Sampling technique for the present study is not present
4- Statistical analysis is not appropriate and does not describe the aim
of the study
5- Scientific soundness for the current study is low
-Calculation of sample size needs to be revised. Which formula or program was used for sample size calculation?
2-Study design and setting need to be thoroughly described
3-Sampling technique for the present study is not present
4- Statistical analysis is not appropriate and does not describe the aim
of the study
5- Scientific soundness for the current study is low
Author Response
Point 1: Calculation of sample size needs to be revised. Which formula or program was used for sample size calculation?
Response 1: The theory behind our power analysis has now been described on lines 140-145, while the statistical software package has now been stated on lines 166-168.
- Point 2: Study design and setting need to be thoroughly described.
Response 2: As already stated in the Study Design section, we performed a cross-sectional study in which all variables were collected at the same time, although the variables related to the pre-pandemic period were inherently retrospective. However, as also suggested by another reviewer, we have now declared the study design also in the title (line 3). We described the setting of the study in the Materials and Methods section (lines 80-84).
- Point 3: Sampling technique for the present study is not present.
Response 3: We have now declared at the very beginning of the Materials and Methods section (line 74) that we enrolled all consecutive patients attending our clinic that met the inclusion criteria of our study.
- Point 4: Statistical analysis is not appropriate and does not describe the aim of the study.
Response 4: The description of statistical methods has been rearranged to better match with the aims stated at the end of the Introduction (lines 146-161), which were also rewritten following the suggestion of another reviewer (lines 68-71). Please note that all the tests used in this work are appropriate for the data being analyzed, and adjustment for differences in age structure between HT and non-HT women was handled by means of ordered logistic regression modelling, which can be seen as the multivariable counterpart of the Mann–Whitney test. Please also note that our power analysis (like all power analyses) was based on a parametric approach, while the tests performed on our data were nonparametric—this actually ensures higher power if sample sizes are not large enough to fully confirm the central limit theorem.
- Point 5: Scientific soundness for the current study is low.
Response 5: In this study, we wanted to focus our attention on post-menopausal women during the first pandemic wave in Italy, one of the first European countries hit by the COVID-19. We understand that the COVID-19 global health emergency has been recently declared over by WHO, but the consequences of this pandemic have been huge and deep, and it is likely that most of them are not yet fully manifest. An analysis of this specific period and its effects on a potentially fragile population like that of postmenopausal women, could help to better understand and potentially to prevent further psychological and physical impairment during a new crisis and cope with possible not yet manifest consequences.
- English very difficult to understand/incomprehensible.
Response 6: We asked a native English speaker to edit our manuscript, as reported in the Acknowledgements.
Round 2
Reviewer 2 Report
Thanks very much. The authors respond to every question raised in the review